# Classification of Contaminated Insulators Using *k*-Nearest Neighbors Based on Computer Vision

**Marcelo Picolotto Corso** [1,*], **Fabio Luis Perez** [1], **Stéfano Frizzo Stefenon** [2], **Kin-Choong Yow** [2], **Raúl García Ovejero** [3] **and Valderi Reis Quietinho Leithardt** [4]

1   Electrical Engineering Graduate Program, Regional University of Blumenau, R. São Paulo 3250 (Itoupava Seca), Blumenau 89030-000, Brazil; fabiotek@furb.br
2   Faculty of Engineering and Applied Science, University of Regina, Regina, SK S4S 0A2, Canada; stefanostefenon@gmail.com (S.F.S.); kin-choong.yow@uregina.ca (K.-C.Y.)
3   Expert Systems and Applications Lab., E.T.S.I.I. of Béjar, University of Salamanca, 37008 Salamanca, Spain; raulovej@usal.es
4   VALORIZA, Research Center for Endogenous Resources Valorization, Instituto Politécnico de Portalegre, 7300-555 Portalegre, Portugal; valderi@ipportalegre.pt
*   Correspondence: marcelopcorso@gmail.com

**Abstract:** Contamination on insulators may increase the surface conductivity of the insulator, and as a consequence, electrical discharges occur more frequently, which can lead to interruptions in a power supply. To maintain reliability in an electrical distribution power system, components that have lost their insulating properties must be replaced. Identifying the components that need maintenance is a difficult task as there are several levels of contamination that are hard to notice during inspections. To improve the quality of inspections, this paper proposes using *k*-nearest neighbors (*k*-NN) to classify the levels of insulator contamination based on images of insulators at various levels of contamination simulated in the laboratory. Computer vision features such as mean, variance, asymmetry, kurtosis, energy, and entropy are used for training the *k*-NN. To assess the robustness of the proposed approach, a statistical analysis and a comparative assessment with well-consolidated algorithms such as decision tree, ensemble subspace, and support vector machine models are presented. The *k*-NN showed up to 85.17% accuracy using the *k*-fold cross-validation method, with an average accuracy higher than 82% for the multi-classification of contamination of insulators, being superior to the compared models.

**Keywords:** classification of insulators; electrical power system; *k*-nearest neighbors; computer vision

## 1. Introduction

Electrical power distribution systems are responsible for providing electricity to consumers [1]. Due to population growth and improved access to energy, it is necessary to have a reliable electricity distribution grid [2]. There is a growing trend towards analyzing electrical system components with the aim of improving efficiency and reducing energy consumption [3,4]. Saline contamination in coastal regions is a problem that must be monitored; if the insulator has greater surface conductivity, it is more likely to develop a failure. The identification of faults in the electrical power system is a difficult task that requires experience from the operator [5].

Researchers have conducted promising studies regarding the evaluation of contamination in insulators installed outdoors using artificial intelligence strategies [6–8]. According to Abouzeid et al. [9], flashover of the insulator occurs when contaminants cover the surface of the insulator, resulting in a reduction in the surface resistance. The leaked current can be used as an evaluation parameter to predict the level of contamination, given by the equivalent salt deposit density (ESDD), on the external surface of the insulators and therefore leaves the system operators alerted to possible failures.

As presented by Soltani and El-Hag [10], an artificial neural network (ANN) in curve fitting can be used to denoise different types of measured signals emitted from partial discharge sources. The ANN can be applied to distinguish electrical discharges in outdoor insulators caused by surface contamination, corona near the insulator surface, and other common types of failure in these components [11]. Defects in insulators give rise to the onset of partial discharges, which has a detrimental effect on the life of the insulator. It is important to identify defective components as early as possible, so appropriate strategies can be implemented [12]. Currently, many fault detection techniques are based on the identification of partial discharges [13], as these may be associated with the presence of faults that are difficult to identify without the use of specific equipment [14].

The application of ANN to assess contamination in insulators can assist in the identification of faults in the electrical network and can prevent shutdown of the electrical power system. The use of an adaptive neuro-fuzzy inference system (ANFIS) is a promising strategy that can stand out from classic models. The ANFIS combined with the wavelet transform can improve the accuracy in the evaluation of contaminated insulators installed outdoors next to rural roads [15].

The use of wavelet transform combined with other algorithms is a promising alternative for the evaluation of noise signals [16]. The data handling (GMDH) model can be used from data filtered by the wavelet transform and outperforms well-established algorithms such as long short-term memory (LSTM) and ANFIS [17]. The results of that analysis were supported by comparisons with wavelet LSTM and wavelet ANFIS, being the fastest GMDH and obtaining equivalent accuracy to the benchmarks. The LSTM is an architecture used in the field of deep learning that has gained interest in forecasting and classification research [18]. The accuracy of the LSTM is high compared with classic approaches; however, the training process requires great computational effort, which increases the training time [19].

Currently, many authors have used computer vision to identify failures in an electrical power system. For example, Nguyen et al. [20] studied power line inspections, and Manninen et al. [21] performed an assessment of the network infrastructure based on computer vision. Some authors have specifically assessed failure conditions in insulators. Shi and Huang [22] presented a strategy for detecting faults in insulators using geometric constraints that can be applied effectively to recognize a damaged insulator. For an accurate classification of the condition, it is necessary to have a large number of images recorded during field inspections. The strategy based on a deep learning model could achieve an accuracy of up to 92.86% when identifying insulators that do not have an insulating surface.

Sampedro et al. [23] presented a system for diagnosing a set of insulators based on deep learning. The analysis was performed using a convolutional neural network (CNN) that aimed to identify the absence of disk insulators installed in series. In this paper, the strategy is based on modeling the similarity between adjacent disks, so it is possible to identify various types of defects from the same model. For robustness in the model, it is necessary to extensively evaluate several video sequences of inspections of the high voltage electrical network.

The evaluation of a lack of insulators in transmission lines is a problem that has been intensively studied using computer vision, and most studies are based on controlled situations in which the lighting conditions and the background of the image are determined. Specific field assessments during inspections of the electrical system are rare due to the scarcity of images with failures. Data augmentation techniques, such as transformation, segmentation, blur, and brightness changes can be a strategy for training the model for field evaluation. According to Tao et al. [24], from a CNN model, using a technique to increase the database, the accuracy of fault detection can reach up to 91% based on a set of standard insulators under various conditions.

Fault detection in insulators can be automated by identifying components with erosion, such as those found in silicone rubber insulators. The use of preprocessing methods for extracting characteristics can improve the classification of failures. In a laboratory analysis

presented by Ibrahim et al. [25], it was possible to identify insulators with this level of erosion following the IEC-60587 standard from a CNN. The application of a deep CNN based on the degree of erosion in silicone rubber showed better results than classical shallow feedforward ANN approaches. The results of the paper show that the proof of concept is valid and can be applied in future tests in an external environment.

An assessment of the impact of contamination caused by rail vehicles on insulators was presented by Kang et al. [26], in which computer vision through a CNN was applied for automatic inspection of the insulators, with the objective of improving the safety of the railway operation. The identification of failures in these locations is difficult due to the low failure rates that exist, which result in few images for the algorithm to be trained on. The anomaly classification is determined from a deep multitasking ANN, which has a material classifier and a denoising autoencoder. The results presented using CNN indicate that the proposed model can achieve high precision for fault identification along a railway line.

Recently, some work has been highlighted for evaluating insulators using [27–29] images, and feature extraction techniques combined with deep learning are efficient for the identification of defects in outdoor insulators [30]. According to Shi and Huang [22], using supervised networks, it is possible to identify missing insulators in transmission lines with an accuracy of up to 92.86%, which can significantly improve inspections of the electrical power system.

The imaging diagnosis of insulators is considerably important since these components are responsible for the support and isolation of electrical energy conductors [31]. Advanced computer vision techniques, such as a ResNeSt, have been increasingly used to identify faults in insulators [32]. The combination of deep convolutional neural network techniques with aerial images was use during inspection results in models that have high accuracy for classifying defects in high voltage networks [33].

One method that has currently stood out for classification and regression is the $k$-nearest neighbors ($k$-NN) [34]. Despite being a method that has been used for several years, many variations of this algorithm are currently being evaluated to improve its capacity. The papers by Mailagaha Kumbure et al. [35] and González et al. [36] combine the $k$-NN algorithm with systems based on fuzzy logic to improve its accuracy. As presented by Sharma and Seal [37], the method used to calculate the distances of this algorithm is an important parameter to be evaluated; for this reason, the distance calculation method is a parameter evaluated in this paper.

The main contributions of this paper are as follows:

- The first contribution of this work is related to the improvement in the diagnosis of contaminated insulators through an artificial intelligence model, which can be used for several applications and shows high efficiency.
- The second contribution is related to computational vision analysis of insulators using non-soluble deposit density. This is an innovative contamination analysis using this measure for electrical power system insulators.
- The third contribution is that the $k$-nearest neighbors model is superior to the decision tree, ensemble, support vector machine, and multilayer perceptron models for this application.

The limitations of the work are due to the difficulty of analysis in the field using contamination evaluation metrics such as non-soluble deposit density and equivalent salt deposit density. As these metrics are the result of a systematic analysis of insulators in the laboratory, field assessments using these measurements are not possible.

The rest of this paper is presented as follows: In Section 2, the problem related to contamination is presented, followed by an explanation of the analysis in the laboratory for the production of the samples and feature extraction. In Section 3, the $k$-nearest neighbors model is presented. In Section 4, the results obtained are discussed and compared with well-established models. Finally, in Section 5, a conclusion is presented with a general discussion about the application of the used algorithm and possible future work.

## 2. Insulator Contamination

Contaminated insulators are a common problem in electrical power distribution and transmission networks. There are several types of contamination, which depend on where the network is installed [38]. Insulators installed near unpaved streets may have an accumulation of dust and organic residues, and insulators that are close to regions with a high population density may have industrial contamination or automobile combustion residues [39–41].

Another very common problem is saline contamination, which is present in coastal regions. These contaminations must be evaluated and monitored by energy utilities, since the salinity combined with an high relative humidity of the air increases the surface conductivity of the insulators and can cause failures in an electrical power system [42].

Common ways to measure the level of contamination in outdoor insulators are the non-soluble deposit density (NSDD) [43] and equivalent salt deposit density (ESDD) [44]. Contamination in coastal regions results in increased surface conductivity of insulators, increasing their leakage current and resulting in discharges that reduce the life of these components [45–47]. With an increase in partial discharges, electrical arcs and flashover can occur [48], which result in carbonization of the contaminants that are embedded in the surface of the insulators [49].

### 2.1. Contaminated Insulator Samples

The insulators used in this paper are 15 kV pin types made of porcelain by the manufacturer Germer. These components meet the criteria of IEC 60383 [50] and are widely used in medium-voltage distribution lines in southern Brazil. The contamination procedure was performed in a high-voltage laboratory at the Regional University of Blumenau, Brazil.

The experimental procedure consists of three stages, which are artificial contamination, photographic capture, and measurement of the level of contamination. To perform the artificial contamination process, the insulators were immersed in a contamination solution based on IEC 60507 [51], which specifically deals with artificial contamination tests on high-voltage ceramic insulators. To create the variations defined by the standard, different contamination conditions were used, and these were obtained from the variation of the mass of Kaolin added to distilled water.

The photographic capture of the insulators under different levels of contamination was carried out with an LG H-818 sensor that has a focal aperture of 1.8. For standardization of photo registration, the exposure time was set at 3.33 ms, light sensitivity was set at ISO 150, and the focal length was set at 4 mm. A photograph of a contaminated insulator is presented in Figure 1.

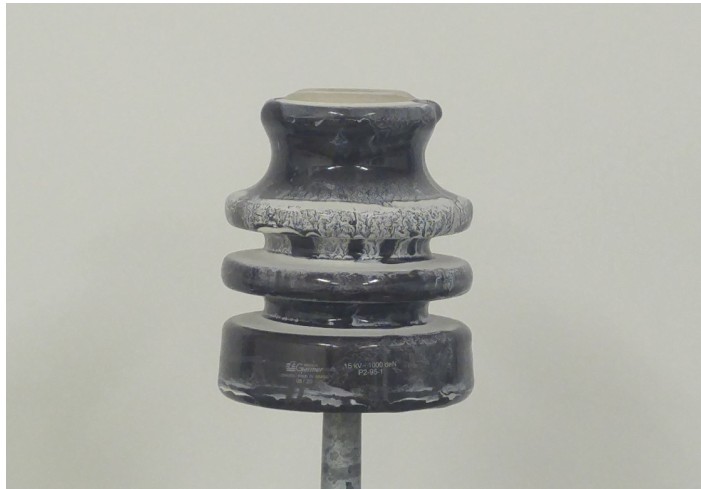

**Figure 1.** Contaminated insulator sample.

The location where the photographs were recorded was standardized to guarantee that the luminous intensity applied to the insulator and that the backgrounds were the same in all photographic captures. The insulators were fixed on a metal pedestal at a distance of 1 m at the same height that the camera was positioned. For each contaminated sample, eight captures are made, two every 90° around the insulator, thus comprising four different viewing directions; the photos on each side were taken using a zoom of 2.5 times and 3.5 times to ensure a greater level of detail.

For determination of the intensity of the NSDD, an IEC 60815 (Annex C) [52] was used, which is specific for the selection and dimensioning of high-voltage insulators under polluted conditions. In this analysis, the residues were extracted from the insulating surface into a container with distilled water and the contents of the container went through a filtering process. The variation in the mass of the filter resulted in the amount of *NSDD* on the insulator, given by the following equation:

$$NSDD = (W_f - W_i)/A, \tag{1}$$

where $W_f$ is the final mass of the filter after the process, $W_i$ is the mass of the filter before the process, and $A$ is the surface area of the insulator, so the value of *NSDD* is given in g/cm$^2$.

For the data set, five different insulators of the same profile were used in eight contamination classes. Thus, a set of data was obtained from 40 conditions of analysis. For comparison, three insulators without contamination were used, resulting in four levels of NSDD for multi-classification. The concentrations of Kaolin used were 6, 8, 10, 16, 20, and 25 (g/l). For two classes, the concentrations of 40 and 60 (S/cm) of salt were used; from these variations, eight classes were obtained. The four levels of NSDD were NSDD equals zero, NSDD greater than 0 and less than 1.0, NSDD from 1.0 to 2.0, and NSDD greater than 2.0.

### 2.2. Image Preprocessing

To perform preprocessing of the image, arithmetic transformations were performed and the first conversion was applied to transform the color image to a grayscale image [53]. In this step, each transformation combined different color channels of the image to result in a grayscale image. After converting the image to grayscale, the image segmentation was applied. Segmentation was performed to partition the image in regions that are based on their characteristics of the image pixels [54].

The segmentation active contours region technique was used to segment the image into foreground and background. Starting from the active contour technique, the initial curves of an image were specified, and then, the active contour function evolved the curves towards the object's limits. Using this segmentation technique, the mask argument is a binary image that specifies the initial state of the active contour [55]. The limits of the object's regions in the mask define the starting position of the contour used for the evolution of the contour to segment the image.

The resulting image is a binary image in which the foreground is white and the background is black. From this conversion resulting in a black background, it is possible to focus on the classification specifically of the insulator, disregarding image noise. Figure 2 presents the result of the conversion and segmentation of the insulator presented in Figure 1.

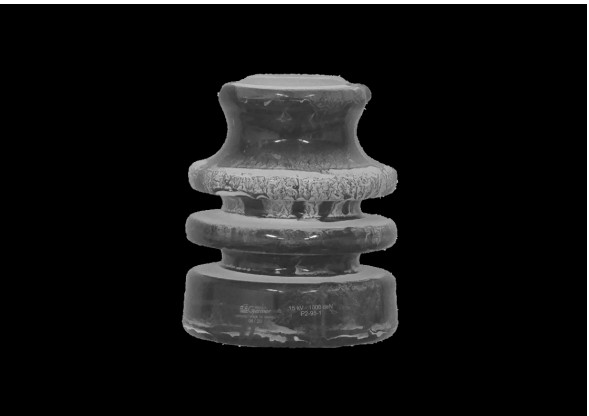

**Figure 2.** Conversion of the photograph of the insulator sample after segmentation.

*2.3. Feature Extraction*

The purpose of the feature extraction is so that the image information is segmented to perform the classification. For this purpose, the information was encoded in a feature vector, given by the following:

$$\vec{x} = [x_1, x_2, x_3, \cdots, x_n]^T, \tag{2}$$

representing the image signature, where $n$ is the total number of attributes considered.

One of the possible ways to extract the features of an image is through histogram $h$ analysis. To obtain the histogram, normalization is performed to obtain the probability density $p$ of the image. This way,

$$p(i) = h(i)/NM, \tag{3}$$

where $i$ is each index of the histogram and $NM$ is the dimensions of the image. From $p$, considering $i \in 0, \cdots, L-1$, wherein $L$ is the total amount of intensity in the image, other measures are obtained, such as the mean (4), variance (5), asymmetry (6), kurtosis (7), energy (8), and entropy (10).

All of the features presented here were used to perform ANN training and testing:

$$\mu_1 = \sum_{i=0}^{L-1} i p(i). \tag{4}$$

$$\mu_2 = \sigma^2 = \sum_{i=0}^{L-1} (i - \mu_1)^2 p(i). \tag{5}$$

$$\mu_3 = \sigma^{-3} \sum_{i=0}^{L-1} (i - \mu_1)^3 p(i). \tag{6}$$

$$\mu_4 = \sigma^{-4} \sum_{i=0}^{L-1} (i - \mu_1)^4 p(i) - 3. \tag{7}$$

$$En = \sum_{i=0}^{L-1} [p(i)]^2. \tag{8}$$

$$Et = -\sum_{i=0}^{L-1} p(i) \log_2 [p(i)]. \tag{9}$$

To make it possible to calculate the entropy logarithm [56] when the value of $p(i)$ is equal to zero, the value of a constant $c$ is added, in this case it is equal to 0.1, and we have the following:

$$Et = -\sum_{i=0}^{L-1} p(i)\log_2[p(i) + c]. \tag{10}$$

The complete dataset with all images and the respective contamination results are available for future comparisons in the Data Availability Statement Subsection at the end of this paper.

## 3. Nearest Neighbors Method

This section presents the *k*-nearest neighbors (*k*-NN) method that was used to classify the insulators evaluated in this paper. The presented method is based on the evaluation of similar data considering the hypothesis of being concentrated in the same region of input space, and non-similar data are distant from each other.

The method of the nearest neighbor has variations defined by the number of neighbors considered; for this reason, the number of neighbors is an evaluation parameter discussed in this paper. In the *k*-NN model, the *k* objects of the training set closest to $x_t$ are evaluated [57]. When *k* is greater than one, the neighboring *k* is obtained for each test point. Figure 3 shows the impact of using the value of *k* on the *k*-NN model. As can be seen, considering $k = 3$, the test is classified with a failed (defective) insulator, while for $k = 5$, the insulator would be classified as a component in good condition.

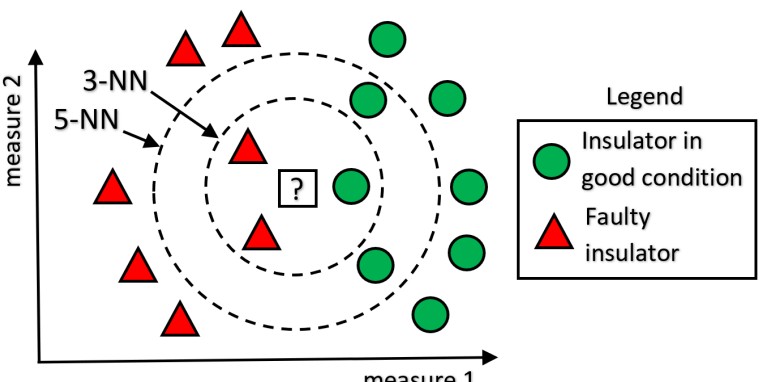

**Figure 3.** Impact of the *k* value on the *k*-NN algorithm.

For this reason, choosing the value of *k* is not a trivial decision and its variation must be evaluated. As it is a classification problem, it is convenient that the value of *k* is odd, thus avoiding draws, which makes classification difficult [58]. For the classification problem, the weighted mode is given by the following:

$$y_t = \arg\max_{c \in Y} \sum_{i=1}^{k} w_i I(c, y_i) \tag{11}$$

wherein,

$$w_i = \frac{1}{d(x_t, x_i)} \tag{12}$$

and $I(a, b)$ is a function that returns 1 if $a = b$ [59]. Since $y_i$ is the class of example $x_i$, $w_i$ is the weight associated with the example of $x_i$ and $c$ is the class with the most weighted mode.

### 3.1. Model Architecture

The *k*-NN is a memory-based algorithm, so all computation is postponed until the classification phase, since the learning process consists of memorizing objects [60]. One of

the advantages of *k*-NN is that the model is simple; thus, it requires less computational effort than methods based on deep learning because, during training, the algorithm only stores objects.

The *k*-NN constructs the approximation of the objective function, which is different for each new data to be stored. This feature can be advantageous when the objective function is complex and can be described as a collection of less complex local approaches [61]. Based on its characteristics, *k*-NN can be applied to complex problems, being an incremental algorithm; that is, when new data are available, it is only necessary to store it in memory.

An important aspect to be considered about the *k*-NN is related to its behavior at the limit if considered:

- *e* = optimal Bayes classifier error;
- $e_{1nn}(D)$: error of 1-NN;
- $e_{knn}(D)$: error of *k*-NN.

Thus, the following theorems are proven:

- $lim_{n \to \infty} e_{1nn}(D) <= 2 \times e$;
- $lim_{n \to \infty, \, k \to n} e_{knn}(D) = e$.

Therefore, for an infinite number of objects, the error of 1-NN is increased by twice the optimal Bayes error, and the error of *k*-NN tends toward the error of the optimal Bayes.

The disadvantages in the *k*-NN are that the model does not obtain a compact representation of the objects, so there is no explicit model from the training data. To classify an object, *k*-NN requires calculating the distance from that object to all training objects. Similar to other algorithms that perform the calculation based on distance, *k*-NN is affected by the presence of redundant attributes and/or irrelevant [62].

One way to improve the capacity of the model is to investigate the reduction in the problem space or to evaluate how the variation of the distance is calculated. For a complete assessment about the *k*-NN, in this paper, many methods to calculate the distance of the neighbors are evaluated, as described in the next section.

### 3.2. Neighbor Distance Method

For a complete evaluation of the variation of the result in relation to the function for the calculation of neighbor distance, in this paper, several functions were used. From an *my*-by-*n* data matrix $Y$, which is treated as my (1-by-*n*) row vectors $y1, y2, ..., y_{my}$, and an *mx*-by-*n* data matrix $X$, which is treated as *mx* (1-by-*n*) row vectors $x1, x2, ..., x_{mx}$, the distances between the vector $x_s$ and $y_t$ could be defined according to the Euclidean distance (13), cosine distance (14), correlation distance (15a), chebychev distance (16), city block distance (17), spearman distance (18), standardized Euclidean distance (19), minkowski distance (20), and mahalanobis distance (21).

$$d_{st}^2 = (x_s - y_t)(x_s - y_t)'. \tag{13}$$

$$d_{st} = \left(1 - \frac{x_s y_t'}{\sqrt{(x_s x_s')(y_t y_t')}}\right). \tag{14}$$

$$d_{st} = 1 - \frac{(x_s - \bar{x}_s)(y_t - \bar{y}_t)'}{\sqrt{(x_s - \bar{x}_s)(x_s - \bar{x}_s)'}\sqrt{(y_t - \bar{y}_t)(y_t - \bar{y}_t)'}} \tag{15a}$$

wherein,

$$\bar{x}_s = \frac{1}{n}\sum_j x_{sj} \tag{15b}$$

and

$$\bar{y}_t = \frac{1}{n} \sum_j y_{tj}. \tag{15c}$$

$$d_{st} = \max_j \left\{ |x_{sj} - y_{tj}| \right\}. \tag{16}$$

$$d_{st} = \sum_{j=1}^{n} |x_{sj} - y_{tj}|. \tag{17}$$

$$d_{st} = 1 - \frac{(r_s - \bar{r}_s)(r_t - \bar{r}_t)'}{\sqrt{(r_s - \bar{r}_s)(r_s - \bar{r}_s)'} \sqrt{(r_t - \bar{r}_t)(r_t - \bar{r}_t)'}}, \tag{18}$$

where $r_s$ and $r_t$ are the coordinate-wise rank vectors of $x_s$ and $y_t$ [63].

$$d_{st}^2 = (x_s - y_t) V^{-1} (x_s - y_t)', \tag{19}$$

wherein $V$ is the $n$-by-$n$ diagonal matrix in which the diagonal element is $S(j)^2$, with $S$ being a vector of scaling factors for each dimension.

$$d_{st} = \sqrt[p]{\sum_{j=1}^{n} |x_{sj} - y_{tj}|^p}. \tag{20}$$

$$d_{st}^2 = (x_s - y_t) C^{-1} (x_s - y_t)', \tag{21}$$

where $C$ is the covariance matrix. Considering that the calculation of the distance from neighbors is of paramount importance for the algorithm, a wide range of functions results in a more complete assessment of the $k$-NN [64].

### 3.3. Holdout

In the holdout approach, the division of data is carried out considering a proportion of data $\rho$ for training and a proportion $(1 - \rho)$ for testing. To make the results less dependent on the partition made, random partitions can be used to obtain an average performance using holdout [65]. For this paper, 80% of the data was used for network training and 20% was used for testing.

### 3.4. Cross-Validation

In the process of cross-validation, the data set was divided into subsets $k$ of approximately equal size. The $k$-1 objects are used in the training of the predictor, which is then tested in the remaining partition. The process is repeated $k$ times, using a different partition in each cycle until all have been used. The predictor's performance are given by the average of the performance observed in each test set [66]. In the classification problem in question, $k$-fold in each partition maintains the proportion of examples from each class similar to the proportion contained in the total data set [67].

### 3.5. Benchmarking

After evaluating the best configuration of the model, a benchmarking with the decision tree [68], ensemble [69–71], support vector machine (SVM) [72–74], and the multilayer perceptron models are presented. The pictures of the insulators were taken before the measurement of the NSDD, so that there was no influence from the operator on the contamination; if the contamination did not meet the requirement of IEC 60815 (Annex C) [52], the process was repeated from the beginning.

The results presented in this paper were evaluated with an Intel Core I5-7400 and 20 GB of random-access memory, with the MATLAB software. The application of the proposed method in an embedded system could be a solution for the electrical inspections,

improving the reliability of the electrical power network [75–77]. In this paper, the accuracy was used given by the coefficient of determination ($R^2$), calculated as follows:

$$R^2 = 1 - \frac{\sum\limits_{i=1}^{n}(y_i - \hat{y}_i)^2}{\sum\limits_{i=1}^{n}(y_i - \bar{y}_i)^2}. \tag{22}$$

where $y$ is the observed value, $\hat{y}$ is the value of the predicted output, and $\bar{y}_i$ is the mean of the targets [78].

### 4. Analysis of Results

In this section, the results of the multi-classification of contaminated insulators are presented and discussed. As the used model is supervised, the first analysis to be performed is related to the NSDD result, which is considered the desired output from the network in the classification process.

Table 1 shows the variation in NSDD depending on the Kaolin concentration used in the analysis. It is noticeable that the NSDD value is directly influenced by the Kaolin concentration used in the experiment. The range of variation of the NSDD can be highm which makes the classification of the data more difficult, since this distribution is not linear. As discussed in Section 2.1, the NSDD value is obtained by the difference of $W_f$ and $W_i$ in relation to the area, according to Equation (1).

**Table 1.** NSDD variation according to Kaolin concentration.

| Kaolin (*g/l*) | 6 | 8 | 10 | 16 | 20 | 25 |
|---|---|---|---|---|---|---|
| Max. $W_f$ | 2.778 | 3.279 | 3.483 | 4.297 | 4.201 | 4.485 |
| Min. $W_f$ | 2.408 | 2.542 | 2.615 | 2.946 | 3.580 | 3.956 |
| Max. $W_i$ | 2.226 | 2.415 | 2.204 | 2.522 | 2.385 | 2.461 |
| Min. $W_i$ | 2.121 | 2.134 | 2.088 | 2.123 | 2.088 | 2.145 |
| Max. NSDD | 0.868 | 1.503 | 1.688 | 2.618 | 2.637 | 2.656 |
| Min. NSDD | 0.377 | 0.393 | 0.633 | 1.029 | 1.568 | 1.661 |

In addition to the six variations in the Kaolin concentration, two classes were analyzed with the inclusion of salt (to evaluate ESDD), resulting in six concentration variations and eight evaluated classes. In Figure 4, the NSDD results are presented in mg/cm$^2$ for the 40 analyses performed, being divided into eight classes (presented in different colors) with five insulators in each class.

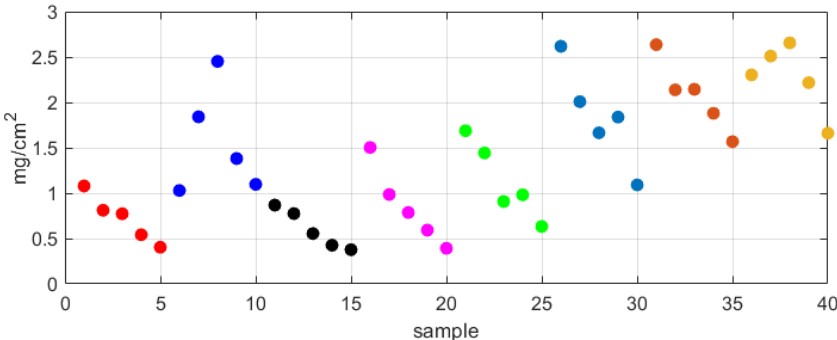

**Figure 4.** Non-soluble deposit density (NSDD) for the samples analyzed.

As can be seen, there is a grouping of data that is related to the levels of NSDD. This result is expected since, for each insulator of the same class, the same concentration of contaminants was used. It was noticed during the initial laboratory evaluation that

ESDD does not generate visual variations; that is, the conductivity that occurs due to the concentration of salt in the analysis does not represent a large visual variation, and for this reason, the ESDD was not evaluated. The NSDD variation caused by the Kaolin concentration generates great visual differences, and for this reason, the NSDD was the focus of the classification analysis.

To perform the multi-classification, the results were evaluated in relation to the NSDD concentration level. Table 2 shows the percentage of insulators that fit each condition evaluated in the multi-classification. In addition, to the 40 samples evaluated, 3 samples without contamination were included for the classification, which were considered to be scale background, so the classification occurs for 4 different conditions.

**Table 2.** Percentage of insulators by contamination level for the classification.

| Kaolin (g/l) | NSDD < 1.0 (mg/cm$^2$) | 1.0 < NSDD < 2.0 (mg/cm$^2$) | NSDD > 2.0 (mg/cm$^2$) |
|---|---|---|---|
| 6 | 100% | 0% | 0% |
| 8 | 80% | 20% | 0% |
| 10 | 60% | 40% | 0% |
| 16 | 0% | 40% | 60% |
| 20 | 0% | 40% | 60% |
| 25 | 0% | 20% | 80% |

After defining the desired output from the network, the classification process begins using the *k*-NN algorithm. The parameters of the algorithm are evaluated dynamically from the variation of the number of neighbors and variation of the function for the calculation of the distance in *k*-NN. To assess the influence of the data separation method, the first analysis was performed with the data directly separated, through the holdout approach, presented in Figure 5.

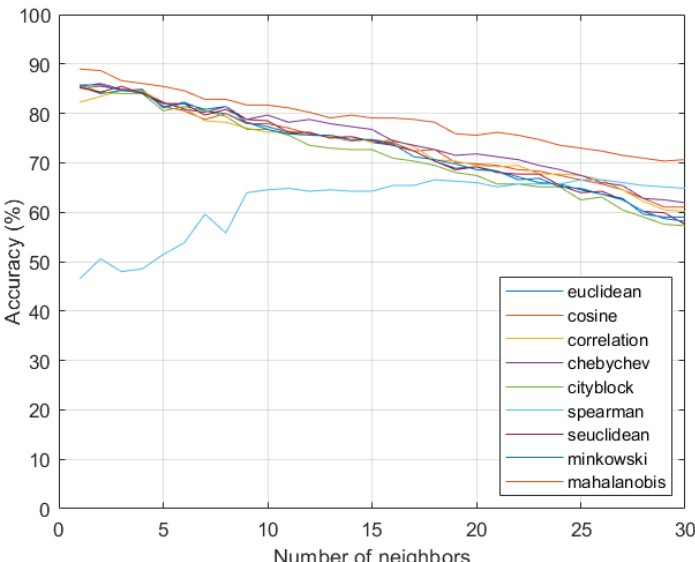

**Figure 5.** Analysis of the accuracy of the algorithm using holdout.

In this initial assessment, as in most methods of distance calculation, the greater the number of neighbors in the *k*-NN, the lower the accuracy of the algorithm. The exception occurs in calculating the distance using the Euclidean equation; however, from 10 neighbors, there is no significant improvement in accuracy through this function. The method of calculation that results in better accuracy for the division of data by the holdout approach is the mahalanobis function. In the following analysis, the evaluation is performed with the validation approach from randomly separated data; these results are shown in Figure 6.

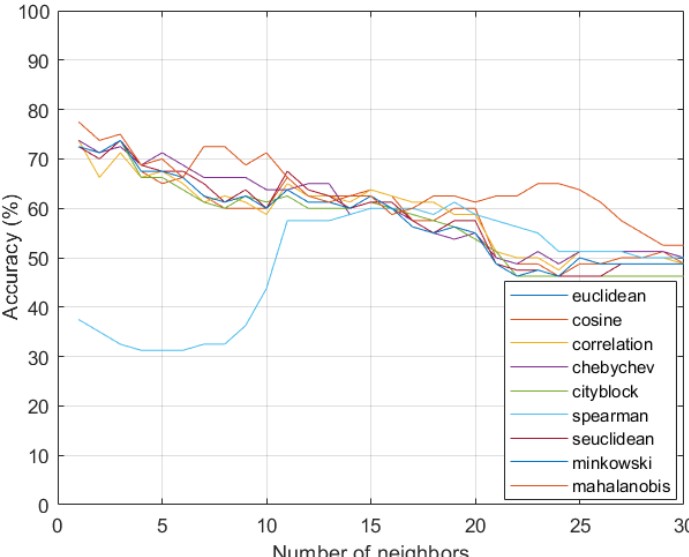

**Figure 6.** Analysis of the accuracy of the model using holdout with randomly separated data.

Using the holdout approach with the data randomly separated, there is a greater variation in accuracy depending on the variation in the number of neighbors. There is also a greater variation using different functions for calculating the distance in the *k*-NN. Despite the greater variation, the characteristics of better accuracy in the mahalanobis function and greater variation of the spearman function as well as those found without the use of randomness remain. Figure 7 presents an analysis of the use of the *k*-fold cross-validation method.

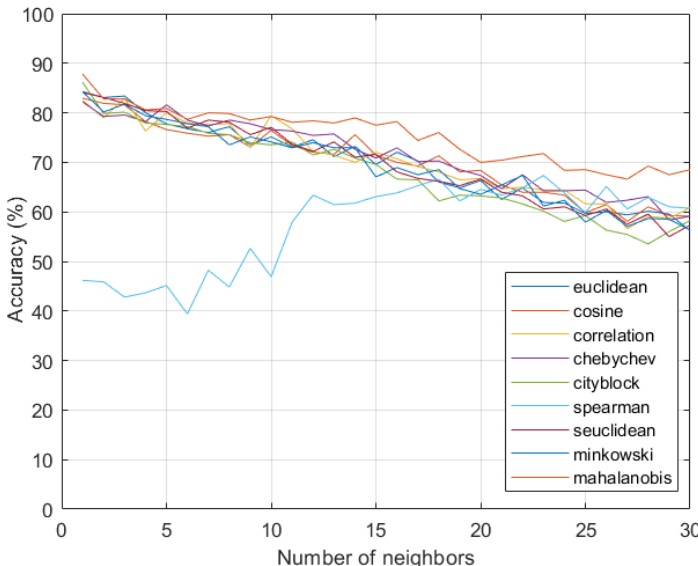

**Figure 7.** Analysis of the accuracy of the algorithm using cross-validation.

As well as the results applying the holdout approach from a random data set, in the *k*-fold method, the best accuracy result is maintained using the mahalanobis function. From the mahalanobis function, there is an improvement in accuracy depending on the way the data set is used in the training process; the best result was obtained using the *k*-fold from two neighbors.

Based on this configuration, the variation in the number of folds (*k*) is evaluated in Table 3 for three distance weight calculation methods. The evaluation was performed using

2 to 12 folds. However, the best results were obtained between 5 to 10; for this reason only these values are presented.

**Table 3.** Evaluation of the variation in the number of folds.

| Distance Weight | Accuracy (%) | | | | | |
|---|---|---|---|---|---|---|
| | **5-Fold** | **6-Fold** | **7-Fold** | **8-Fold** | **9-Fold** | **10-Fold** |
| Equal | 82.85 | 81.69 | 82.85 | 81.69 | 82.85 | 79.94 |
| Inverse | 79.36 | 81.98 | 82.85 | 80.52 | **84.59** | 81.69 |
| Sq. Inver. | 81.69 | 84.30 | 82.85 | 83.14 | 84.58 | 83.14 |

The best accuracy result was found using nine folds in the three methods of calculating the distance weight. By using this configuration, 100 analyses were performed, and the statistical result is shown in Table 4.

**Table 4.** Statistical evaluation of the *k*-NN algorithm.

| Distance Weight | Accuracy (%) | | | Std. Dev. |
|---|---|---|---|---|
| | **Max.** | **Min.** | **Mean** | |
| Equal | **85.17** | 79.65 | 82.39 | $8.70 \times 10^{-3}$ |
| Inverse | **85.17** | 77.91 | 82.26 | $1.11 \times 10^{-2}$ |
| Sq. Inverse | 84.88 | 79.36 | 82.23 | $9.6 \times 10^{-3}$ |

The best accuracy found was 85.17 %, for calculating the distance weight from 100 analyzes. It is possible to realize that this variation is statistically low based on the values of standard deviation; this shows that the algorithm is robust with low variation even with a large number of analyzes.

*Benchmarking*

For a comparative evaluation, the same configuration used for the cross-validation of the *k*-NN was applied in all compared models, and the statistical results are presented in Table 5. The model variations are presented for a complete assessment. The decision tree algorithm uses split criterion Gini's diversity index (gdi) and deviance. In SVM, the coding "one vs. one" and "all pairs" are applied.

**Table 5.** Benchmarking with statistical evaluation.

| Method | Accuracy (%) | | | Std. Dev. |
|---|---|---|---|---|
| | **Max.** | **Min.** | **Mean** | |
| Decision Tree (gdi) | 56.98 | 47.09 | 52.92 | $1.26 \times 10^{-2}$ |
| Decision Tree (deviance) | 61.63 | 56.40 | 60.02 | $7.43 \times 10^{-3}$ |
| Ensemble (subspace) | 67.44 | 63.95 | 65.53 | $6.42 \times 10^{-3}$ |
| SVM (onevsone) | 47.67 | 43.31 | 45.51 | $6.34 \times 10^{-3}$ |
| SVM (allpairs) | 47.38 | 42.73 | 45.40 | $7.84 \times 10^{-3}$ |
| Multilayer perceptron | **76.25** | 66.25 | 70.87 | $2.27 \times 10^{-2}$ |

As can be seen in Table 5, the decision tree, ensemble (subspace), support vector machine, and multilayer perceptron models had lower accuracy results than the *k*-NN for the evaluation presented in this paper. The differences between the maximum and minimum values remained low in all evaluated algorithms, considering 100 simulations with the same configuration using the cross-validation of the data. The best result found in benchmarking was using the multilayer perceptron, which resulted in an accuracy of 76.25% in the best case.

## 5. Conclusions

The use of the *k*-NN presented in this paper demonstrated that it is possible to classify contamination in insulators as a promising accuracy. Based on this classification, it is possible to define strategies for maintaining the distribution network according to the levels of contamination, which are found on the surface of the insulating components. Through predictive maintenance, it is possible to improve the reliability of the network by reducing power outages.

The analysis showed that cross-validation, a method covered in several articles, is superior to the holdout method for this application. The best accuracy results were found from a lower number of neighbors. This shows that it is necessary to make a complete evaluation of the algorithm to find its best configuration, which in consequence results in higher accuracy. The functions for calculating the distance from neighbors obtained results that followed the same trend, with the exception of the Euclidean function, which had an inverse result in relation to the other functions. The most accurate function for calculating neighbors was the mahalanobis for all validation methods evaluated.

From the best configuration found for the model, the statistical analysis showed a low variation, considering that 100 analyzes were performed using random weights. For all distance calculation methods, the variation between the worst result and the best result was less than 10%, and in the best case, the accuracy reached 85.17%. This result was superior to that of well-consolidated algorithms such as decision tree, ensemble, support vector machine, and multilayer perceptron. Based on the promising results found in this paper, it is possible in the future to carry out analyzes to classify the conditions of the insulators in the field. Other methods of extracting characteristics can be applied to compare the influence of the characteristics on the classification results. In the future, registration of the photos may be performed using drones, facilitating visualization and classification of the conditions of the insulators through algorithms based on artificial intelligence as well as those presented in this paper.

**Author Contributions:** Conceptualization, M.P.C. and S.F.S.; methodology, S.F.S.; software, M.P.C.; validation, S.F.S.; formal analysis, M.P.C. and S.F.S.; investigation, M.P.C.; resources, F.L.P.; writing—original draft preparation, M.P.C.; writing—review and editing, S.F.S.; supervision, F.L.P., K.-C.Y., R.G.O., and V.R.Q.L.; funding acquisition, R.G.O. and V.R.Q.L. All authors have read and agreed to the published version of the manuscript.

**Funding:** This work was supported by national funds through the Fundação para a Ciência e a Tecnologia, I.P. (Portuguese Foundation for Science and Technology) by the project UIDB/05064/2020 (VALORIZA—Research Centre for Endogenous Resource Valorization). Al Proyeto: Uso de algoritmos y protocolos de comunicación en dispositivos con énfasis en la privacidad de los datos.

**Institutional Review Board Statement:** Not applicable.

**Informed Consent Statement:** Not applicable.

**Data Availability Statement:** For future comparisons, the data and the *k*-NN algorithm are available at https://github.com/SFStefenon/InsulatorsDataSet (accessed on 8 September 2021).

**Acknowledgments:** The authors thank the Emerging Leaders in the Americas Program (ELAP), Canadian Bureau for International Education (CBIE), Government of Canada, who provided a scholarship for visiting graduate research students at the University of Regina, Canada. The authors are thankful to the Coordination for the Improvement of Higher Education Personnel (CAPES) for the Master's scholarship awarded to Marcelo P. Corso.

**Conflicts of Interest:** The authors declare no conflicts of interest.

**Abbreviations**

The following abbreviations are used in this manuscript:

| | |
|---|---|
| ANFIS | adaptive neuro-fuzzy inference system |
| ANN | artificial neural network |
| CAPES | Coordination for the Improvement of Higher Education Personnel |
| CBIE | Canadian Bureau for International Education |
| CNN | convolutional neural network |
| ELAP | Emerging Leaders in the Americas Program |
| ESDD | equivalent salt deposit density |
| GMDH | group method of data handling |
| $k$-NN | $k$-nearest neighbors |
| LSTM | long short term memory |
| NSDD | non-soluble deposit density |
| SVM | support vector machine |

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
