# Peer review of "Classification of Contaminated Insulators Using k-Nearest Neighbors Based on Computer Vision"

_computers, doi:10.3390/computers10090112_

Round 1

Reviewer 1 Report

Classification of Contaminated Insulators using k-Nearest Neighbors based on Computer Vision

This research work proposes computer vision based approach to classify insulator contamination. To validate the proposed approach, different experiments are performed. Furthermore, a comparative analysis is performed to verify the performance of proposed approach over other common classification techniques. Overall, the paper is easy to be understood and results are presented in a clear way.

In general, the title and abstract are appropriate; the subject fits with the journal’s aims and scope. the introduction is clear, the review of literature is well researched and proper. The flow/ different blocks of proposed scheme is well defined.

The research is quite interesting for readers, especially researchers and developers that are working on real-time computer vision applications.

Only a few suggestions will be proposed in the following, to further improve the paper:

Literature review should also discuss the traditional approaches (like Visible Light Images, other hyper-spectral approaches etc) to measure the insulator contamination, few more references need to be added.

The proposed approach should also be compared with other approaches like Multilayer Perceptron Networks and random forest for a complete analysis.

Author Response

This research work proposes computer vision based approach to classify insulator contamination. To validate the proposed approach, different experiments are performed. Furthermore, a comparative analysis is performed to verify the performance of proposed approach over other common classification techniques. Overall, the paper is easy to be understood and results are presented in a clear way.

Thank you for taking the time to read your paper and for your suggestions for improvements.

In general, the title and abstract are appropriate; the subject fits with the journal’s aims and scope. the introduction is clear, the review of literature is well researched and proper. The flow/ different blocks of proposed scheme is well defined.
The research is quite interesting for readers, especially researchers and developers that are working on real-time computer vision applications.

We appreciate your comment, we are happy that you liked our paper.

Only a few suggestions will be proposed in the following, to further improve the paper:

Literature review should also discuss the traditional approaches (like Visible Light Images, other hyper-spectral approaches etc) to measure the insulator contamination, few more references need to be added.

We appreciate the suggestion to improve the literature review, to meet this suggestion we have included the following text:

Recently, some works have been highlighted for evaluating insulators using [A-C] images, feature extraction techniques combined with deep learning are efficient for the identification of defects in outdoors insulators [D]. According to Shi and Huang [E], using supervised networks it is possible to identify missing insulators in Transmission lines with an accuracy of up to 92.86 %, which can significantly improve inspections of the electrical power system.

The imaging diagnosis of insulators is considerably important since these components are responsible for the support and isolation of electrical energy conductors [F]. Advanced computer vision techniques, such as a ResNeSt, have been increasingly used to identify faults in insulators [G]. The combination of deep convolutional neural network techniques with aerial images taken during inspection results in models that have high accuracy for classifying defects in high voltage networks [H].

Common ways to measure the level of contamination in outdoor insulators are the non-soluble deposit density (NSDD) [I] and equivalent salt deposit density (ESDD) [J]. Contamination in coastal regions results in increased surface conductivity of insulators, increasing their leakage current and resulting in discharges that reduce the life of these components [K-M]. With an increase in partial discharges, electrical arcs and flashover occur [N], which result in carbonization of the contaminants that are embedded in the surface of the insulators [O].

The proposed approach should also be compared with other approaches like Multilayer Perceptron Networks and random forest for a complete analysis.

As per your suggestion, we included the comparison using the multilayer perceptron (Table 5), we chose to use the subspace (ensemble) model to compare with the ensemble class, which is a model similar to a random forest.

References:

[A] Han,  Y.;  Liu,  Z.;  Lee,  D.;  Liu,  W.;  Chen,  J.; Han,  Z.    Computer  vision–based  automatic  rod-insulator  defect  detection in  high-speed  railway catenary  system. International  Journal  of  Advanced  Robotic  Systems 2018, 15,  1-15. 
[B] Yin, J.; Lu, Y.; Gong, Z.; Jiang, Y.; Yao, J. Edge Detection of High-Voltage Porcelain Insulators in Infrared Image Using Dual Parity Morphological Gradients. IEEE Access 2019, 7, 32728–32734.
[C] Kokalis,  C.C.A.;   Tasakos,  T.;   Kontargyri,  V.T.;   Siolas,  G.;   Gonos,  I.F. Hydrophobicity  classification  of  composite  insulators  based  on  convolutional  neural  networks. Engineering  Applications  of  Artificial  Intelligence 2020, 91,  103613.
[D] Wen, Q.; Luo, Z.; Chen, R.; Yang, Y.; Li, G. Deep Learning Approaches on Defect Detection in High Resolution Aerial Images of Insulators. Sensors 2021, 21, 1033.
[E] Shi, C.; Huang, Y. Cap-Count Guided Weakly Supervised Insulator Cap Missing Detection in Aerial Images. IEEE Sensors Journal 2021, 21, 685–691.
[F] Ma, Y.; Li, Q.; Chu, L.; Zhou, Y.; Xu, C. Real-Time Detection and Spatial Localization of Insulators for UAV Inspection Based on Binocular Stereo Vision. Remote Sensing 2021, 13.
[G] Wang, S.; Liu, Y.; Qing, Y.; Wang, C.; Lan, T.; Yao, R. Detection of Insulator Defects With Improved ResNeSt and Region Proposal Network. IEEE Access 2020, 8, 184841–184850.
[H] Waleed, D.; Mukhopadhyay, S.; Tariq, U.; El-Hag, A.H. Drone-Based Ceramic Insulators Condition Monitoring. IEEE Transactions on Instrumentation and Measurement 2021, 70, 1–12.
[I] Maadjoudj, D.; Mekhaldi, A.; Teguar, M. Flashover process and leakage current characteristics of insulator model under desert pollution. IEEE Transactions on Dielectrics and Electrical Insulation 2018, 25, 2296–2304.
[J] Wardman, J.; Wilson, T.; Bodger, P. Volcanic ash contamination: limitations of the standard ESDD method for classifying pollution severity.IEEE Transactions on Dielectrics and Electrical Insulation 2013, 20, 414–420. 
[K] Maraaba, L.S.; Soufi, K.Y.A.; Alhems, L.M.; Hassan, M.A. Performance Evaluation of 230 kV Polymer Insulators in the Coastal Area of Saudi Arabia.IEEE Access 2020, 8, 164292–164303. 
[L] Su, H.; Jia, Z.; Guan, Z.; Li, L. Mechanism of contaminant accumulation and flashover of insulator in heavily polluted coastal area. IEEE Transactions on Dielectrics and Electrical Insulation 2010, 17, 1635–1641.  
 [M] Salem, A.A.; Abd-Rahman, R.; Al-Gailani, S.A.; Salam, Z.; Kamarudin, M.S.; Zainuddin, H.; Yousof, M.F.M. Risk Assessment of Polluted Glass Insulator Using Leakage Current Index Under Different Operating Conditions. IEEE Access 2020, 8, 175827–175839.
[N] Matsuoka, R.; Kondo, K.; Naito, K.; Ishii, M.  Influence of nonsoluble contaminants on the flashover voltages of artificially contaminated insulators.IEEE Transactions on Power Delivery 1996, 11, 420–430. 
[O] Jiang, X.; Yuan, J.; Zhang, Z.; Hu, J.; Sun, C. Study on AC Artificial-Contaminated Flashover Performance of Various Types of Insulators. IEEE Transactions on Power Delivery 2007, 22, 2567–2574. 

Reviewer 2 Report

In this manuscript, the authors propose a new approach for classification of contaminated insulators by using a combination from k-nearest neighbors machine learning method and computer vision techniques.
The manuscript is well-structured.
The research goal is clearly formulated.
The analysis of experimental results is thorough.

My main concerns about the proposed manuscript are as follows:
In the abstract and "Introduction" section, please highlight the novelty of the study in detail. What are your contributions?
MATLAB code/pseudo-code could be added.
In the experimental section, only a simulated-dataset is employed. Link to this dataset is missing.
Please, apply your approach to existing real-life contaminated insulators dataset/s.
In section "4. Analysis of Results", a comparison with results from similar studies is missing.
Please, describe what the limitations of your study are.

Technical remark:
l. 40: The reference number is missing.

Author Response

In this manuscript, the authors propose a new approach for classification of contaminated insulators by using a combination from k-nearest neighbors machine learning method and computer vision techniques. The manuscript is well-structured. The research goal is clearly formulated. The analysis of experimental results is thorough.

We appreciate your comments and your review.

My main concerns about the proposed manuscript are as follows:
In the abstract and "Introduction" section, please highlight the novelty of the study in detail. What are your contributions?

As per your suggestion, we include contributions:

The main contributions of this paper are:

* The first contribution of this work is related to the improvement of the diagnosis of contaminated insulators through an artificial intelligence model, which can be used for several applications and shows high efficiency.
* The second contribution is related to computational vision analysis of insulators using non-soluble deposit density. This being an innovative contamination analysis using this measure for electrical power system insulators.
* The third contribution is that the \textit{k}-nearest neighbors model is superior to the decision tree, ensemble, support vector machine, and multilayer perceptron models for this application.

MATLAB code/pseudo-code could be added.

Excellent suggestion, we included the code in Matlab along with the database.

This information is at: https://github.com/SFStefenon/InsulatorsDataSet.

In the experimental section, only a simulated-dataset is employed. Link to this dataset is missing.

We have included the following text in the experimental section:

The complete dataset with all images and the respective contamination results are available for future comparisons in the Data Availability Statement subSection at the end of this paper.

Please, apply your approach to existing real-life contaminated insulators dataset/s.

The purpose of this paper is to evaluate the influence of contamination in relation to the classification of non-soluble deposit density (NSDD).
To present an analysis using the NSDD it is necessary to produce samples in the laboratory, as there is a defined standard for how to carry out the contamination. For this reason, we do not use insulators taken from the field. We consider the suggestion to be very promising when the analysis is not carried out in relation to the NSDD. To discuss the classification of real-life contaminated insulators, we seek to present a discussion of how other authors performed classification using machine learning.

In section "4. Analysis of Results", a comparison with results from similar studies is missing.

This suggestion is very important to be discussed. There are no works that use NSDD based on computer vision for the classification of contamination. Typically this metric is related to direct analysis, such as leakage current analysis. For this reason, we do not include comparisons with other authors in the analysis section. We include a comparative analysis through benchmarking to show that the algorithm used is suitable for the problem at hand.

Please, describe what the limitations of your study are.

As per your suggestion, we have included the following text to talk about the limitations of the study:

The limitations of the work are due to the difficulty of analysis in the field using contamination evaluation metrics such as non-soluble deposit density and equivalent salt deposit density. As these metrics are the result of a systematic analysis of insulators in the laboratory, field assessments using these measurements are not possible.

Technical remark:
l. 40: The reference number is missing.

We've included a reference, thank you for the correction.
